# Pricing Decisions on Online Channel Entry for Complementary Products in a Dominant Retailer Supply Chain

**Qiongqiong Gu** [1] , **Xiaodong Yang** [2] **and Bin Liu** [1,*]

1   School of Economics and Management, Shanghai Maritime University, Shanghai 201306, China; guqiongqiong0012@stu.shmtu.edu.cn
2   College of International Trade, Shanghai University of International Business and Economics, Shanghai 201620, China; sheldon.yang123@gmail.com
*   Correspondence: liubin@shmtu.edu.cn

**Abstract:** This study considered the supply chain that two manufacturers sell green complementary products to a dominant offline retailer. We investigated whether a manufacturer (the integrated manufacturer) should add an online channel and examined how it affects channel members' decisions and profits. We formulated the power structure as the retailer-Stackelberg model and analyzed the pricing decisions for the supply chain. The results demonstrate that the integrated manufacturer prefers not to add the online channel when online and offline market bases are comparable and the level of complementarity is moderate. The integrated manufacturer gains more power at the expense of the offline retailer and the other manufacturer (the traditional manufacturer) when the complementarity between the offline and online channel is the same as offline channels with the addition of a new online channel; furthermore, the retailer earns less, while the traditional manufacturer's profit hinges on the complementarity between the online and offline channels. It is beneficial for the offline retailer to balance the online and offline market bases of product 1 by improving the sales environment of the physical store. The integrated manufacturer can benefit from varying their marketing actions to decrease the degree of complementarity between the retail and online channels for the two products, while the traditional manufacturer can be better off from the online channel introduction by taking steps to increase the complementarity of the two products between the offline channels.

**Keywords:** green complementary products; online channel; pricing; supply chain; Stackelberg game

## 1. Introduction

Recently, electronic commerce is growing at an unprecedented rate with the popularization of the Internet. More and more manufacturers compete with retailers by multi channels, consisting of indirect channels and direct channels [1–3]. Direct channel operations have been added by a large number of manufacturers: Nike, IBM, Hewlett-Packard, and Eastman Kodak, for example [4]. In such instances, the direct channel of the manufacturer competes with the traditional channel of the retailer; as a result, the traditional retailer ordinarily complains that the demands that are satisfied through the manufacturer's direct channel should belong to the retailer's traditional channel [5]. This may result in the channel conflict, which weakens the attempts to build the cooperative relationship for channel members [6]. Sales through the manufacturer's direct channel dramatically cannibalize the market share and profits of the traditional retailer. Some operational strategies have been adopted (such as providing services and value-added products) by a lot of traditional retailers to

strengthen their core competitiveness. As reported by the New York Times, there are roughly 42% of manufacturers reconstructing the traditional channel structures through adding direct online sales to satisfy different customer segments that are not reachable by the traditional channel, which are generating the combination channel of the traditional channel and booming the direct online channel (also called the dual-channel). The proliferation of the direct supply channel has evoked concerns about manufacturer encroachment as a hot topic of discussion [7]. However, not all studies have found that the addition of direct channel creates conflicts, some researchers have shown that the addition of a new channel is beneficial to other channel members. For example, Arya et al. [8] demonstrated the bright side of manufacturer encroachment. Therefore, on this basis, we study the interesting issues of whether a manufacturer opens a new direct channel in the supply chain, where green complementary products are sold, and how other channel members adopt strategies when adding an online channel into the traditional channel.

Green production has gradually become a popular and growing field due to its important role in economic and environmental sustainability. Some researchers focus on the issue of green product to reduce environment damage while maintaining profits [9]. Now, as the laws and regulations become more and more rigorous, the organizations cannot ignore product green level. For example, the company Apple, as core designer of the product, sets the green product target, then for satisfying the target its suppliers will increase the recycling of manufacturing material and reduce the use of toxic metal. This paper mainly considers the sales of green complementary products.

The meaning of complementary products refers to when customers must buy above one product to acquire the products' entire utility at the same time [10], such as camera and film, memory card and mobile phone, pencils and erasers. Unlike substitutable products, the feature about the complementary products is that the sales of the products benefit from sales of each other [11]. In that case, the demands for manufacturers' products are interlinked in the same market. Market performance of one manufacturer would be affected by another manufacturer's marketing decision.

The green complementary products refer to environmentally friendly complementary products, for example, a practical case of green products is about the house decoration business—when Noritz sells an energy-saving water heater, Chicago Furnace also provides environmentally friendly faucets with water-saving outlet options (Contractor Magazine. Oct 2017, Vol. 64 Issue 10, p43.). Therefore, the manufacturers of green complementary products may confront the problem of channel selection and pricing in the supply chain.

Previously, the manufacturers always are powerful in a supply chain. The power of supply chain members lies in their ability to affect the decision variables of other members [12]. The last several decades, however, have seen the emergence of powerful retailers (also known as "dominant retailers") like Best Buy, Walmart, and Home Depot. The rise of dominant retailers has aroused discussions about their effects on supply chain members, since dominant retailers can be dominating and tough when dealing with manufacturers [13,14].

In this paper, we study a supply chain which consists of a dominant offline retailer and two manufacturers (integrated manufacturer and traditional manufacturer), who sell the retailers green complementary products. The integrated manufacturer represents the manufacturer that can sell the product through two channels: the direct online channel and traditional offline channel, while the traditional manufacturer means that the manufacturer only sells product through the traditional offline channel. We examine whether the integrated manufacturer should add a new online channel and how the online entry would affect other channel members. The issue is addressed by the following questions:

Under what circumstances should the integrated manufacturer in the traditional channel add the online channel?

1.  How should channel members make their decisions in the online and offline channels?
2.  How does the online expansion of the integrated manufacturer in a traditional channel affect the channel members' prices, demands, and profits?

This paper involves three streams of research: manufacturer encroachment, dual-channel/multi-channel supply chains, and complementary products. We briefly review the literature in the following three areas.

The effects of manufacturer encroachment have been widely discussed. Most of the literature indicates that manufacturer encroachment harms retailers through intensifying the channel competition [15,16]; however, some scholars show that manufacturer encroachment is beneficial to both the manufacturer and retailer. Chiang et al. [5] illustrated that Pareto gains occur as the manufacturer adds a direct channel. Arya et al. [8] demonstrated the bright side of manufacturer encroachment, which benefits the retailer from online channel entry. More recently, Guan et al. [17] investigated the manufacturer's voluntary disclosure strategy; selling to the end consumers directly can infringe on the retailer's business. Zhang and Li [18] demonstrated the effect of asymmetric demand information and endogenous quality decision on manufacturer encroachment. Unlike the above studies, our paper studies manufacturer encroachment in the context of complementary products rather than substitute products in the supply chain.

The dual-channel supply chain literature considers mainly the scenario where all the supply chain members offer substitute products or a single product. A significant amount of study about dual-channel supply chains has concentrated on pricing decisions of substitute products. Chen et al. [19] studied single period pricing strategies of substitute products in supply chain. Yao and Liu [20] demonstrated that the addition of online channels about a single product causes competitive prices and payoffs. Cai [21] evaluated the impact of pricing schemes with a single product and price discount contracts on competition in the dual-channel supply chain. Kurata et al. [22] studied the pricing decisions of substitute products in the supply chain, where there is a contest between the store brand and the national brand. Batarfia et al. [23] studied the theme that the manufacturer sells the consumers the customized products by the online channel and sells them the pre-configured products directly in a dual-channel. However, only a few studies have focused on the pricing problems of complementary products in dual-channel supply chains.

Much of the literature has researched complementary products from different perspectives. Li et al. [24] researched whether the policy of markdown pricing in the sales cycle is inferior to bundling complementary products. Yue et al. [11] and Wei et al. [25] studied the information sharing of two duopoly companies providing complementary products. Only a few papers have investigated the pricing decisions of complementary products [26–29], but they either focused on channel members' different powers regarding the complementary products in the traditional channel [26], or considered the power structure of supply chains [30–32]. Our analysis is most closely related to Zhao et al. [28] and Wang et al. [29], who both studied the complementary products in the dual-channel supply chain. Wang et al. [29] considered both the service and price decisions of complementary products with varying supply chain structures, while Zhao et al. [28] derived the optimal pricing strategies for complementary products under different power structures. Our paper departs from these studies in that we consider manufacturers' online entry in the existing traditional channel for complementary products.

The two most relevant studies studied the impact of different market powers and two types of channel pricing forms on supply chain profits by formulating four game models. However, in all these studies, the manufacturer was modeled as the leader in the market. Our work differs in part by exploring whether the powerful retailer influences integrated manufacturers' decisions on their online entry. Our contributions to this paper are four-fold. First, we demonstrate under what conditions the integrated manufacturer in the traditional channel should add an online channel. Second, how the added online channel affects the demands, prices, and profits of channel members is discussed. Third, supply chain members should strive to make the degree of complementarity between online and offline for the two products lower than that between the offline channels, which will benefit the supply chain members. Fourth, when the online and offline market bases are almost the same and the level of complementarity is moderate, the integrated manufacturer does not add the online channel, and the traditional supply chain may bring more revenue.

The remainder of this paper is arranged as follows. The problem statement and the model description are presented in Section 2. The equilibrium analyses are conducted and feasibility conditions are discussed in Section 3. Section 4 compares the profits and strategies across models, and determines the effect of the online channel entry on channel members. In Section 5, we have some extensions on the complementarity between the online and traditional channels. Finally, Section 6 concludes the implications of the findings.

## 2. Notation and Model

This study considered a supply chain consists of two manufacturers (denoted as *the integrated manufacturer* and *the traditional manufacturer*) and a retailer (*she*), who sell two complementary products. The product $i$ is produced through the manufacturer $i$(*he*) with unit cost $c_i$, then they wholesale the product $i$ to the offline retailer with unit price $w_i$, next the retailer resells the product $i$ to final consumers with retail price $p_i$, where $c_i < w_i < p_i$, $i = 1, 2$. The product 1 and product 2 are complementary to each other. The decision variables are chosen by the retailer and the two manufacturers for maximizing their own profits. We further suppose that the two manufacturers and the retailer are risk neutral. Manufacturer 1 is now wondering whether they should add a new online channel to sell product 1 directly, in order to improve their profits. In this paper, Manufacturer 1, who sells product 1 by dual channel, is called the integrated manufacturer, and Manufacturer 2, who sells product 2 only via traditional channel, is called the traditional manufacturer.

The retailer-Stackelberg game represents the rise of the dominant retailer, which is the result of the transfer of the power from other channel members to the retailer. Some retailers, for example, Walmart, Home Depot, and Best Buy, can boost product sales by reducing price, and meanwhile maintaining their own marginal profits by squeezing profits from suppliers (manufacturers). In this paper, we constructed the channel structure model for the sequential noncooperative game, in which the Stackelberg leader is the offline retailer, while the followers are the integrated manufacturer and the traditional manufacturer. In addition, both manufacturers own the same power in the game, that is, the two manufacturers move simultaneously.

We considered two modeling scenarios. The first was the benchmark scenario, where the integrated manufacturer does not add a new online channel, and we call it *Scenario N*, the second was the case that the integrated manufacturer adds an online channel, and we denote it *Scenario E*. The channel structures of the two scenarios are described in Figure 1.

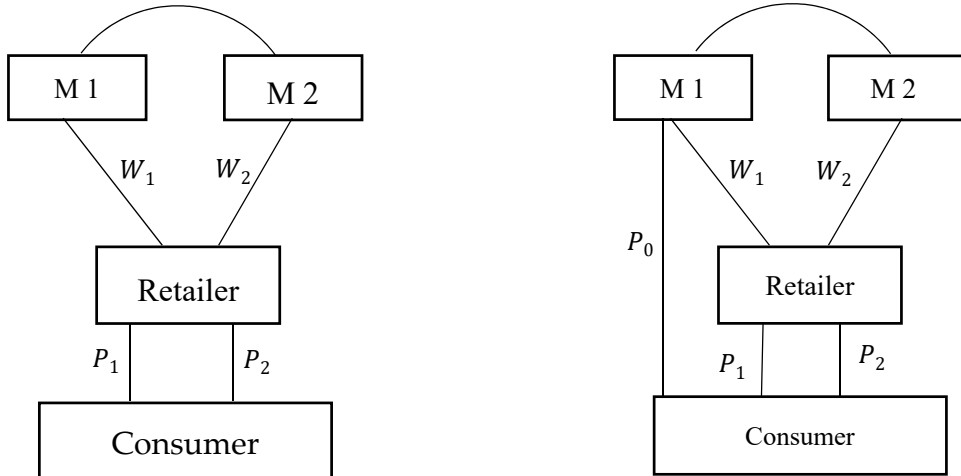

**Figure 1.** Supply chain framework.

### 2.1. The Benchmark Model (Scenario N)

In this section, the benchmark scenario that the integrated manufacturer does not add a new online channel is analyzed. The two complementary products are only sold to the dominant offline retailer from the two manufacturers. Let $D_1$ and $D_2$ denote the demand for product 1 and product 2, respectively, then we have:

$$D_1^N = a_1 - p_1 - \gamma_1 p_2, \tag{1}$$

$$D_2^N = a_2 - p_2 - \gamma_1 p_1, \tag{2}$$

where the parameters $a_1$, $a_2$ denote the base demand of the product 1 and product 2, respectively, the parameter $\gamma_1$ represents the degree of complementarity between product 1 and product 2 in traditional channels, also known as the cross-price sensitivity. It is clear that $0 < \gamma_1 < 1$. For simplicity, own price sensitivities for both products are assumed to be identical and normalized to 1. We assume all other parameters are nonnegative. Own price sensitivities are assumed to be greater than the cross-price sensitivity, because the products' demands are often more sensitive to the change in their prices than to those of their complementary products.

With the above notation, the profits of the offline retailer, traditional manufacturer, and integrated manufacturer can be written by:

$$\pi_{m1}^N(w_1) = (w_1 - c_1)D_1^N, \tag{3}$$

$$\pi_{m2}^N(w_2) = (w_2 - c_2)D_2^N, \tag{4}$$

$$\pi_r^N(p_1, p_2) = (p_1 - w_1)D_1^N + (p_2 - w_2)D_2^N. \tag{5}$$

### 2.2. Dual-Channel Model (Scenario E)

In this model, the integrated manufacturer sells product 1 directly via the added online channel and at the same time continues to sell through the offline retailer to reach the end customers. The traditional manufacturer only sells product 2 to the offline retailer, who resell it in turn to the final customers. The decision variables of the retailer and the traditional manufacturer remain unchanged, but the integrated manufacturer needs to set a direct-channel sales price for his product. $p_0$ denotes the direct-channel sales price for product 1. The unit cost of the integrated manufacturer's direct sales to consumers is normalized to $\delta$. When the integrated manufacturer adds an online channel, product 1 can be purchased by consumers either offline from the retailer or online directly from the integrated manufacturer. The demand functions are given, respectively, by:

$$D_0^E = \theta a_1 - p_0 + \beta p_1 - \gamma_0 p_2, \tag{6}$$

$$D_1^E = (1 - \theta)a_1 - p_1 + \beta p_0 - \gamma_1 p_2, \tag{7}$$

$$D_2^E = a_2 - p_2 - \gamma_0 p_0 - \gamma_1 p_1. \tag{8}$$

Most of the assumptions discussed in the previous section remain unchanged. The parameter $\theta \in (0, 1)$ represents the online base in the market, which is the percentage of base demand for the online channel. It captures the scale of the online base demand, that is, consumer willingness to buy product 1 online. The parameter $\theta$ also indicates product 1's compatibility in the new online channel (Yan, 2011). The market of product 1 is segmented according to channel usage, and hence the offline market base $(1 - \theta)$ increases as the online market base decreases. Parameter $\beta (0 < \beta < 1)$ represents the competition degree between the offline channel and online channel, which is seen as a measurement of the ability of product demand to respond to changes in its price under different distribution channels. The price sensitivities for each of these three channels are assumed to be normalized to 1, it means own price sensitivities are assumed to be greater than the cross-price sensitivities, as mentioned before. The parameter $\gamma_0 (0 < \gamma_0 < 1)$ denotes the degree of complementarity existing between product 1

of the new online channel and product 2, and $\gamma_1 (0 < \gamma_1 < 1)$ denotes the level of complementarity existing between two products in the offline channels.

Based on the above notations, the expressions of the profits for all supply chain members are obtained as follows:

$$\pi_{m1}^E(w_1, p_0) = (w_1 - c_1)D_1^E + (p_0 - c_1 - \delta)D_0^E, \tag{9}$$

$$\pi_{m2}^E(w_2) = (w_2 - c_2)D_2^E, \tag{10}$$

$$\pi_r^E(p_1, p_2) = (p_1 - w_1)D_1^E + (p_2 - w_2)D_2^E. \tag{11}$$

## 3. Equilibrium Analysis

### 3.1. The Benchmark Model (Scenario N)

The timing of the retailer-Stackelberg game is that the retailer moves first and sets the retail prices for two complementary products in the offline channels; next, the two manufacturers set the wholesale prices for two products simultaneously. The game can be constructed as:

$$\begin{cases} \max_{(p_1, p_2)} \pi_r(p_1, p_2, w_1^*(p_1, p_2), w_2^*(p_1, p_2)) \\ w_1^*(p_1, p_2), w_2^*(p_1, p_2) \text{ are derived from solving the following problem} \\ \quad \begin{cases} \max_{w_1} \pi_{m1}(w_1) \\ \max_{w_2} \pi_{m2}(w_2) \end{cases} \end{cases}. \tag{12}$$

The parameter $m_i$ represents the marginal revenue of product $i$ obtained by the retailer, and then we have:

$$p_i = m_i + w_i, \tag{13}$$

where $m_i > 0, i = 1, 2$.

The game is solved backwards. In Scenario N, given the decision of $p_1$ and $p_2$ made by the retailer, the optimal wholesale prices of the integrated manufacturer, and the traditional manufacturer can be given, respectively, as:

$$w_1^{N*}(p_1, p_2) = a_1 + c_1 - p_1 - \gamma_1 p_2, \tag{14}$$

$$w_2^{N*}(p_1, p_2) = a_2 + c_2 - p_2 - \gamma_1 p_1. \tag{15}$$

Substituting $w_1^{N*}(p_1, p_2)$ and $w_2^{N*}(p_1, p_2)$ from Equations (14) and (15) into the Equation (16) and performing the maximization of Equation (5), then the optimal solutions of $p_1$ and $p_2$ are obtained:

$$p_1^{N*} = A_1/A, p_2^{N*} = A_2/A, \tag{16}$$

where where $A, A_1, A_2$ are defined in Appendix A.

The expressions of the equilibrium prices $w_i^{N*}(i = 1, 2)$ of the manufacturer $i$, as provided in Proposition 1, are obtained through the above result.

**Proposition 1.** *In Scenario N, the optimal wholesale prices $w_1^{N*}$ and $w_2^{N*}$ of the complementary products can be given as:*

$$w_1^{N*} = a_1 + c_1 - p_1^{N*} - \gamma_1 p_2^{N*}, \tag{17}$$

$$w_2^{N*} = a_2 + c_2 - p_2^{N*} - \gamma_2 p_1^{N*}. \tag{18}$$

*The proof of Proposition 1 and other remaining proofs appear in Appendix B. According to the above equations, we can obtain the results of Corollary 1.*

**Corollary 1.** *With some simple algebraic manipulations, the sensitivity of the optimal prices about the parameters* $a_1, a_2$ *and* $c_1, c_2$ *are obtained as:*

$$(a) \frac{\delta p_i^{N*}}{\delta a_i} = \frac{6 - 3\gamma_1^2}{2\gamma_1^4 - 10\gamma_1^2 + 8} > 0, \frac{\delta p_i^{N*}}{\delta c_i} = \frac{2 - 2\gamma_1^2}{2\gamma_1^4 - 10\gamma_1^2 + 8} > 0,$$

$$(b) \frac{\delta p_1^{N*}}{\delta a_2} = \frac{\delta p_2^{N*}}{\delta a_1} = \frac{2\gamma_1^3 - 5\gamma_1}{2\gamma_1^4 - 10\gamma_1^2 + 8} < 0, \frac{\delta p_1^{N*}}{\delta c_2} = \frac{\delta p_2^{N*}}{\delta c_1} = \frac{\gamma_1^3 - \gamma_1}{2\gamma_1^4 - 10\gamma_1^2 + 8} < 0,$$

$$(c) \frac{\delta w_i^{N*}}{\delta a_i} = \frac{-2}{2\gamma_1^2 - 8} > 0, \frac{\delta w_i^{N*}}{\delta c_i} = \frac{\gamma_1^2 - 6}{2\gamma_1^2 - 8} > 0,$$

$$(d) \frac{\delta w_1^{N*}}{\delta a_2} = \frac{\delta w_2^{N*}}{\delta a_1} = \frac{\delta w_1^{N*}}{\delta c_2} = \frac{\delta w_2^{N*}}{\delta c_1} = \frac{\gamma_1}{2\gamma_1^2 - 8} < 0.$$

(19)

With the increase of market base for the products $i (i = 1, 2)$, the retail price and wholesale price of product $i$ increase, while those of product $j (j = 3 - i)$ decrease. When the manufacturing cost of the product $i (i = 1, 2)$ increases, the retail price and wholesale price of product $i$ increase, while those of product $j (j = 3 - i)$ decrease. These results are consistent with the findings in early studies, such as Wei et al. [26].

It is very hard to analytically describe the changes of optimal solutions in terms of the complementarity in Scenario N. Hence, numerical analyses were conducted to demonstrate the effects of the complementarity level on the wholesale price, retail price, and demand of product 1. The results are plotted in Figures 2 and 3. In the numerical studies, we set $a_1 = a_2 = 300, c_1 = c_2 = 20$, and $\gamma_1 \in \{0.3, 0.4, 0.5, 0.6, 0.7\}$. Due to the symmetry, products 1 and product 2 had the same retail prices and wholesale prices.

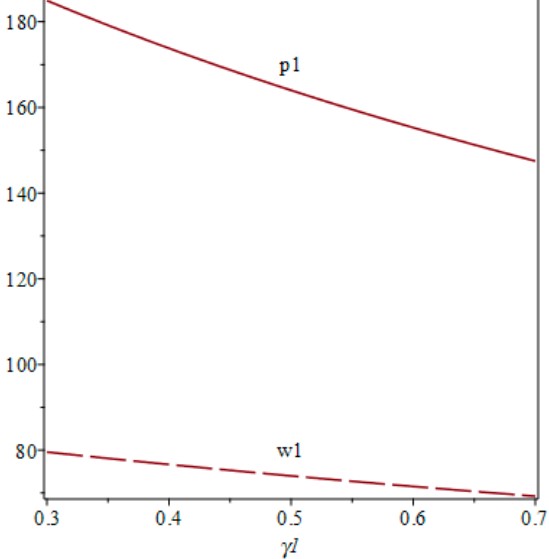

**Figure 2.** Optimal prices versus $\gamma_1$ in Scenario N.

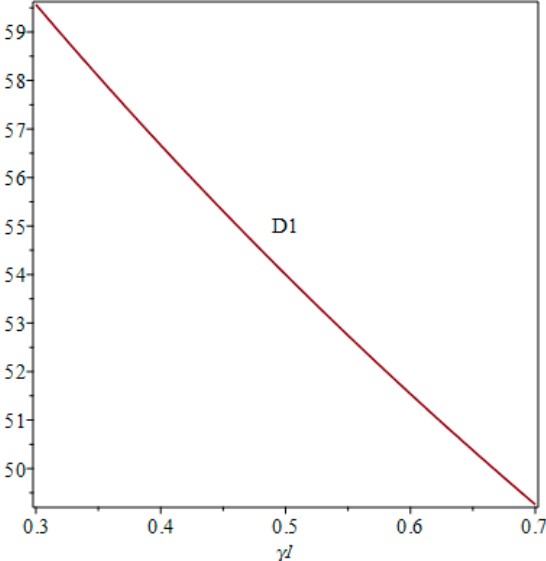

**Figure 3.** Maximum demands versus $\gamma_1$ in Scenario N.

As illustrated in Figures 2 and 3, in Scenario N, with an increase in the complementary level $\gamma_1$, the equilibrium retail prices $(p_1^{N*}, p_2^{N*})$ and wholesale prices $(w_1^{N*}, w_2^{N*})$ decrease. A change in the complementarity level has the similar impact on the demands of product 1 and product 2, as a result, the equilibrium profits of the retailer and the two manufacturers decrease as the level of complementary $\gamma_1$ increases.

### 3.2. Dual Channel Model (Scenario E)

In Scenario E, the dominant retailer again moves first and sets the retail prices for product 1 and product 2; then, the integrated manufacturer sets the online price $p_0$ and the wholesale price $w_1$ simultaneously, and the traditional manufacturer chooses their wholesale price $w_2$ at the same time. The timing of game is shown as:

$$
\begin{cases}
\max\limits_{(p_1,p_2)} \pi_r(p_1,p_2,w_1^*(p_1,p_2),w_2^*(p_1,p_2),p_0^*(p_1,p_2)) \\
w_1^*(p_1,p_2),w_2^*(p_1,p_2),p_0^*(p_1,p_2) \text{ are derived from} \\
\quad \begin{cases} \max\limits_{(w_1,p_0)} \pi_{m1}(w_1,p_0) \\ \max\limits_{w_2} \pi_{m2}(w_2) \end{cases}
\end{cases}
. \qquad (20)
$$

**Proposition 2.** *In Scenario E, first the retailer makes the decisions $p_1$ and $p_2$, then the optimal direct channel price $p_0^{E*}(p_1,p_2)$ and the optimal wholesale prices $w_1^{E*}(p_1,p_2), w_2^{E*}(p_1,p_2)$ of the duopolistic manufacturers are obtained, respectively, as:*

$$w_1^{E*}(p_1,p_2) = B_{11}p_1 + B_{12}p_2 + B_{13},$$

$$p_0^*(p_1,p_2) = B_{21}p_1 + B_{22}p_2 + B_{23}, \qquad (21)$$

$$w_2^{E*}(p_1,p_2) = B_{31}p_1 + B_{32}p_2 + B_{33}.$$

**Corollary 2.** $\dfrac{\delta p_0^*(p_1,p_2)}{\delta p_1} = B_{21} = 0.$

Some managerial implications from Corollary 2 can be inferred: product 1's online price is irrelevant to its offline price decided by the offline retailer. The main reason for the phenomenon is retailer-dominated; the retailer first sets the margin profit of product 1. From Equation (A12), we find the online price of product 1 is positively correlated with its wholesale price, meanwhile positively correlated with the offline price-margin profit of product 1, from Equation (A11), we find a negative correlation between the wholesale price and the offline price, the margin profit of product 1, so the change in the wholesale price decided by the integrated manufacturer as an offset to the change of the offline price, therefore, the wholesale price is a mediator in the supply chain with dominant retailer.

Substituting Equation (21) into profit function of the offline retailer in Equation (11) to maximize her profits, the results are obtained in Proposition 3.

**Proposition 3.** *In Scenario E, the equilibrium retail prices $p_1^{E*}$ and $p_2^{E*}$ can be obtained, respectively, as:*

$$
\begin{pmatrix} p_1^{E*} \\ p_2^{E*} \end{pmatrix} = \begin{pmatrix} c_{11} & c_{12} \\ c_{21} & c_{22} \end{pmatrix}^{-1} \begin{pmatrix} c_{13} \\ c_{23} \end{pmatrix}.
\tag{22}
$$

Consequently, Proposition 4 can be derived easily from Propositions 2 and 3.

**Propositionk 4.** *The two manufacturers' optimal wholesale prices and the online channel sales price of product 1 in Scenario E are:*

$$
w_1^{E*} = B_{11} p_1^{E*} + B_{12} p_2^{E*} + B_{13},
$$

$$
p_0^* = B_{21} p_1^{E*} + B_{22} p_2^{E*} + B_{23},
\tag{23}
$$

$$
w_2^{E*} = B_{31} p_1^{E*} + B_{32} p_2^{E*} + B_{33}.
$$

In Scenario E, the decision variables are set by channel members for the purpose of maximizing their own profits. To ensure channel members' participation (e.g., nonnegative demand, margins, and profits), the parameters need to meet certain conditions. We summarize these conditions in Table 1.

**Table 1.** Notations and positive pricing decisions and margins some assumptions.

| | |
|---|---|
| $D_1^E$ | Demand Offline of Product 1, $D_1^E > 0$ |
| $D_2^E$ | Demand of product 2, $D_2^E > 0$ |
| $D_0$ | Demand online of product 1, $D_0 > 0$ |
| $w_i^E$ | Wholesale price, $w_i^E > c_i$ |
| $p_1^E$ | Price offline of product 1, $p_1^E > w_1^E$ |
| $p_0$ | Price online of product 1, $p_0 > w_1^E$ |
| $p_2^E$ | Price offline of product 2, $p_2^E > w_2^E$ |

In Scenario E, it is less straightforward to analyze the feasibility conditions of the equilibrium problem. Hence, a numerical study was used to explore the feasibility conditions of the problem. For illustration purposes, $\gamma_0$ was set equal to $\gamma_1$, that is, the level of complementarity between two products is same, no matter which channel Product 1 is distributed through. Furthermore, we set $a_1 = a_2 = 300$, $c_1 = c_2 = 20$, $\delta = 2$.

The optimization problem in Scenario E is not feasible when $\theta$ is sufficiently small, that is, $\theta \in (0, 0.22)$, it cannot reach the constrained condition: $p_0 > w_1^E$ , or when $\theta$ is bigger than 0.93, it cannot reach the constrained condition: $D_1^E > 0$. For any $\theta \in (0.22, 0.93)$, not all values of parameters can be verified, only certain parameter values satisfy the conditions in Table 1. Figure 4 shows the feasible domain and infeasible domain (IF) for the optimization problem in terms of $\gamma_1$ and $\beta$ when parameter

$\theta$ varies at 0.35, 0.5, and 0.7. For example, $\gamma 1(\beta)$ and $\gamma 2(\beta)$ represent the border line of $p_0 > w_1^E$ and $D_2^E > w_2^E$ when $\theta = 0.35$ in Figure 4, respectively.

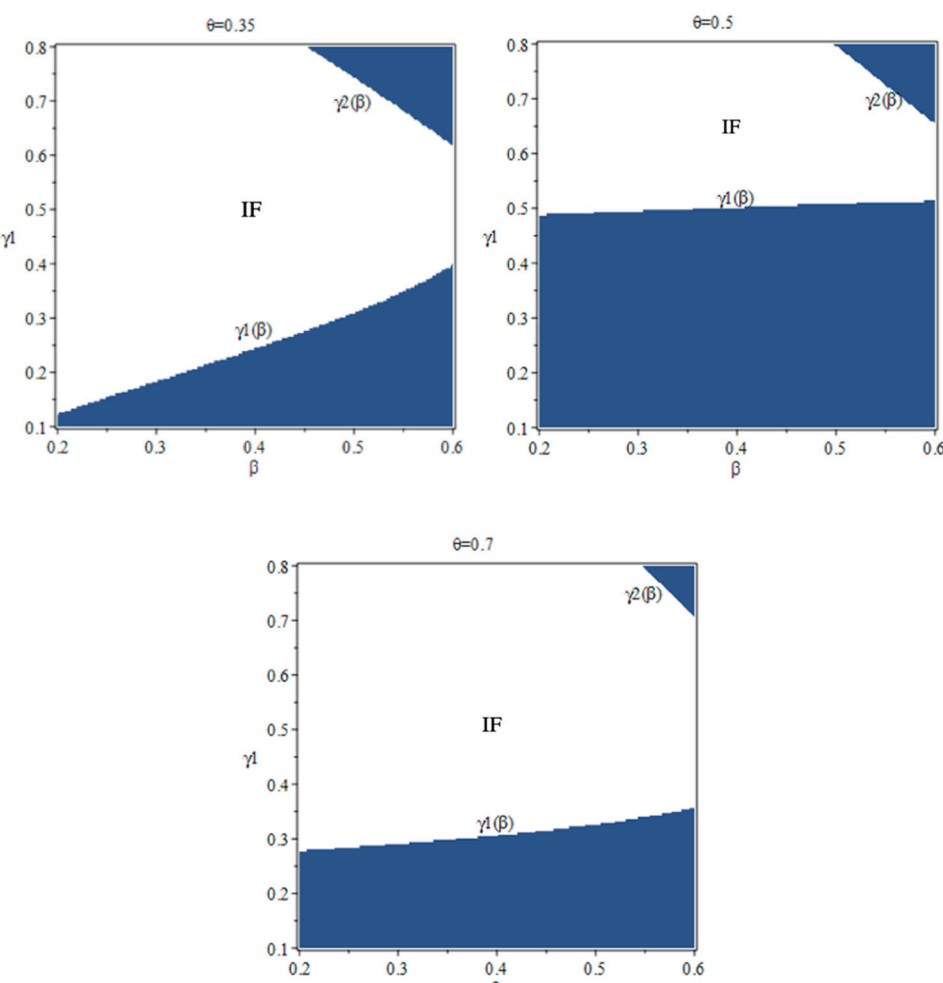

**Figure 4.** Feasible domain for the problem in Scenario E.

As shown in Figure 4, we can see that Scenario E is likely to occur when $\gamma_1$ is either sufficiently large or small. When the competition between the offline and online channels becomes fiercer (i.e., as $\beta$ increases), Scenario E is more likely to be sustained. Moreover, when $\theta \in (0.22, 0.5)$, the area of feasible region increases as the portion of online base demand increases, while the opposite is true for $\theta \in (0.5, 0.93)$. To better understand the feasible domain for the problem in scenario E, let us take a look from a different perspective. As shown in Figure 4, the problem is not feasible for $\gamma_1 \in (\gamma_1(\beta), \gamma_2(\beta))$. The infeasible region sustains for a wide range of $\gamma_1$ when the portion of online base demand is relatively small or large.

## 4. Effects of the Introduction of Online Channel

Now, we are ready to compare the prices, demands, and profits of channel members between Scenarios N and E and discuss the impact of adding the online channel. Due to the complexity of equilibrium solutions, it is very hard to obtain any meaningful insights through comparing the equilibriums analytically, and therefore numerical study was adopted again. We set $a_1 = a_2 = 300$, $c_1 = c_2 = 20$, $\delta = 2$ and changed the values of $\beta$, $\gamma_1$ and $\theta$ in the following ranges: $\beta \in (0.2, 0.6)$, $\gamma_1 \in (0.1, 0.8)$, and $\theta \in (0.22, 0.93)$ in the feasible areas identified in Figure 4. All parameter values

were also verified in the benchmark model. The discussion is presented below. In the following figures, the sign "+" ("−") represents a positive (negative) impact of adding online channel. The curves between the signs represent the boundaries where the benchmark model and the addition of online model generated the same solutions on channel prices, demands, or profits.

### 4.1. On Channel prices

We compared three channel members' prices between the benchmark and dual-channel models and summarize the results in the following claims.

**Claim 1.** *The offline retailer may set a higher or lower retail price for product 1 as the integrated manufacturer adds the online channel, depending on the market base ($\theta$), the competition between the online and traditional offline channels ($\beta$), and the degree of complementarity between the two products in the offline channels ($\gamma_1$).*

*Claim 1 indicates that, depending on model parameters, the retailer may set a higher or lower retail price for product 1 as the integrated manufacturer adds the online channel. As shown in Figure 5, in the case of low online market base (i.e., low $\theta$), product 1 of the dual-channel is likely to have a higher offline price than that of the benchmark model in the conditions of strong competition intensity. Otherwise, when the online market base is moderate or high, the product 1's offline price is usually lower in the dual-channel than that in the benchmark model unless both the competition intensity ($\beta$) and the complementary level ($\gamma_1$) are very large. In general, the retailer is more likely to decide a lower retail price for product 1 in the dual-channel model as the online market base ($\theta$) increases.*

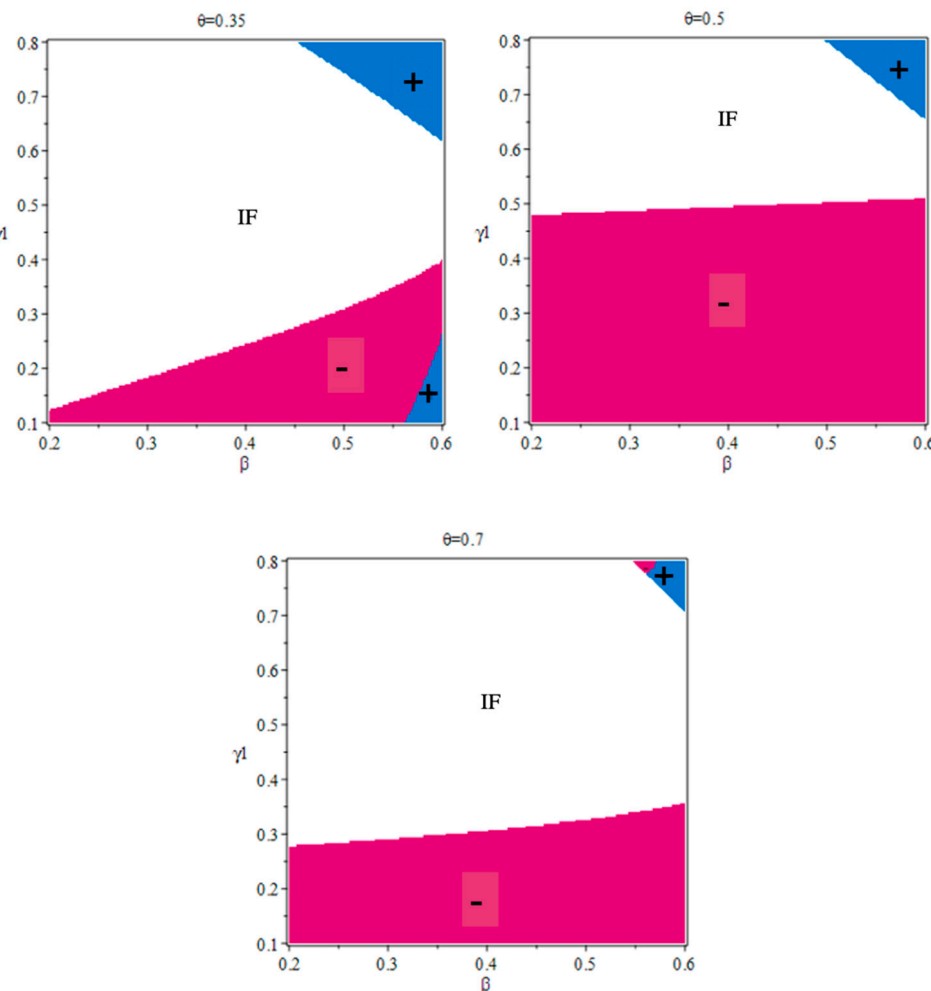

**Figure 5.** Compare the retailer's prices of product 1 (price in scenario E–price in scenario N).

**Claim 2.** *The retailer may set a higher or lower retail price for product 2 as the integrated manufacturer adds the online channel, depending on the market base ($\theta$), the competition between the online channel and traditional offline channel ($\beta$), and the degree of complementarity between the two products ($\gamma_1$).*

Similarly, Claim 2 also demonstrates that the retailer may charge a higher or lower offline price for product 2 as the integrated manufacturer adds the online channel. As shown in Figure 6, when the degree of complementarity ($\gamma_1$) between two offline channels is sufficiently high, the retailer charges a lower retail price for product 2 in Scenario E. In general, the product 2's retail price can be higher in Scenario E than that in Scenario N when the competition intensity ($\beta$) and the online market base ($\theta$) between the offline and online channels are both relatively low. Moreover, the retail price of product 2 is more likely to be higher in Scenario E as the online market base ($\theta$) increases.

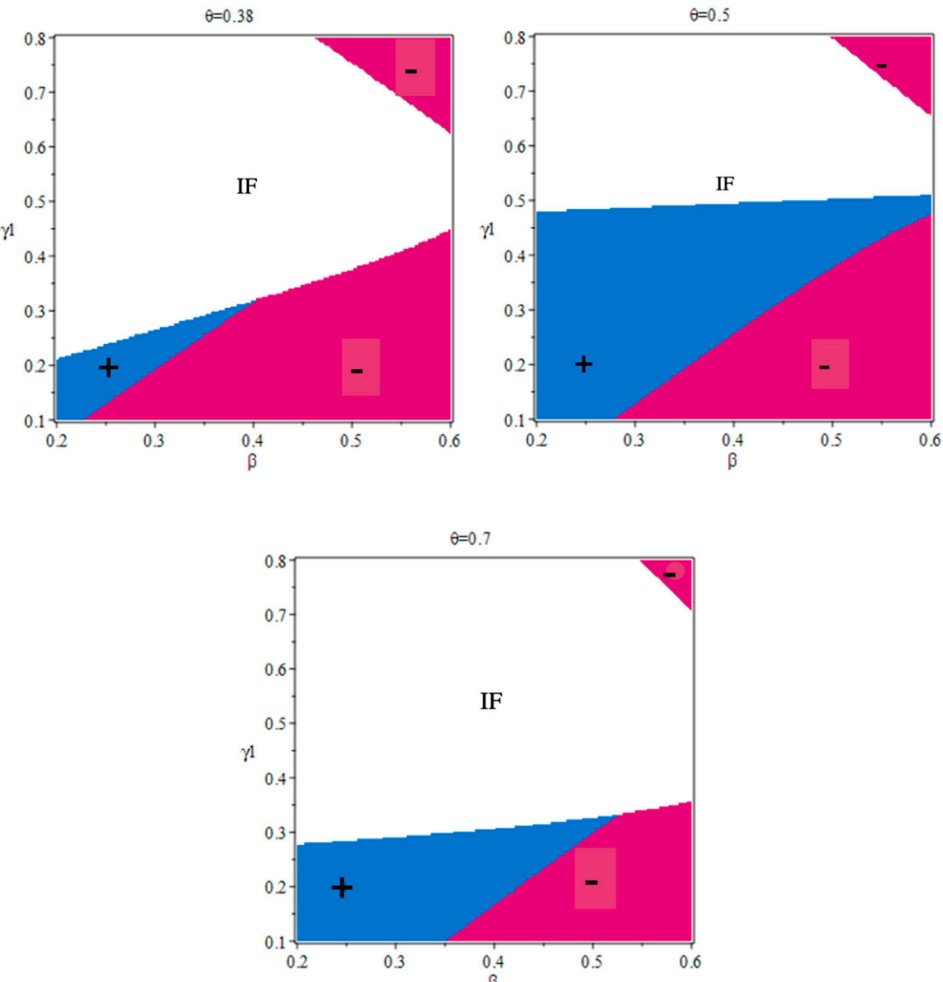

**Figure 6.** Compare the retailer's prices of product 2 (price in scenario E–price in scenario N).

**Claim 3.** *The integrated manufacturer may charge a higher or lower wholesale price of product 1 when adding an online channel, depending on the competition intensity between the online and traditional offline channels ($\beta$), the degree of complementarity between the two products ($\gamma_1$), and the market base ($\theta$).*

From Claim 3, we find that the integrated manufacturer may charge a lower or higher wholesale price of product 1 when adding a new online channel, depending on the variance of the model parameters. Product 1's wholesale price is higher in Scenario E than that in Scenario N when the competition intensity ($\beta$) is sufficiently large, as shown in Figure 7. Otherwise, product 1's wholesale price is likely to be lower in scenario E when the degree of complementarity ($\gamma_1$) between two offline channels is fairly low. In general, the integrated

*manufacturer charges a lower wholesale price for product 1 in the dual-channel model as the online market base ($\theta$) increases.*

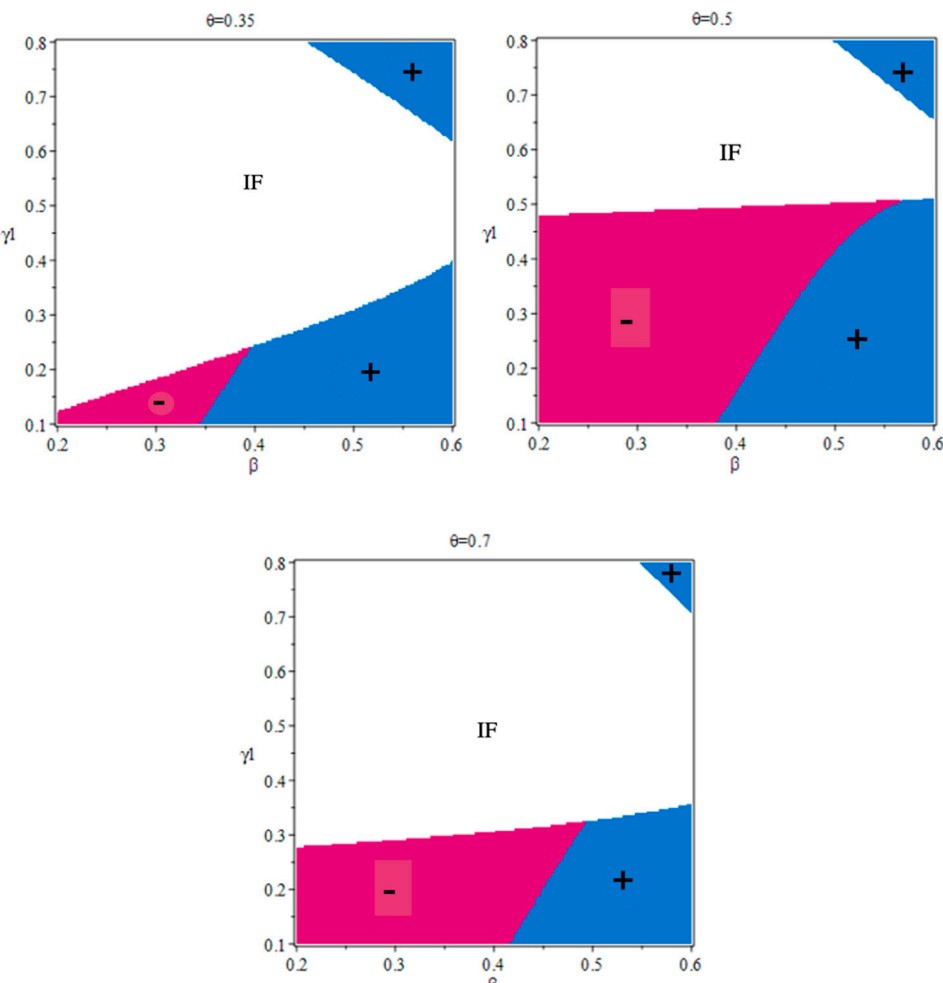

**Figure 7.** Compare integrated manufacturer's wholesale prices (price in scenario E–price in scenario N).

**Claim 4.** *A lower wholesale price of product 2 is charged through the traditional manufacturer as the integrated manufacturer adds the online channel.*

**Claim 5.** *In Scenario E, product 1 of the online channel has a lower retail price than that in the offline channel when $\theta \in (0.22, 0.5)$. Otherwise, when $\theta \in (0.5, 0.62)$, the online retail price can be either lower or higher, depending on the complementarity level ($\gamma_1$) and the competition intensity ($\beta$). Furthermore, when $\theta \in (0.62, 0.93)$, the retail price for product 1 is higher in the online channel than that in the offline channel (Figure 8).*

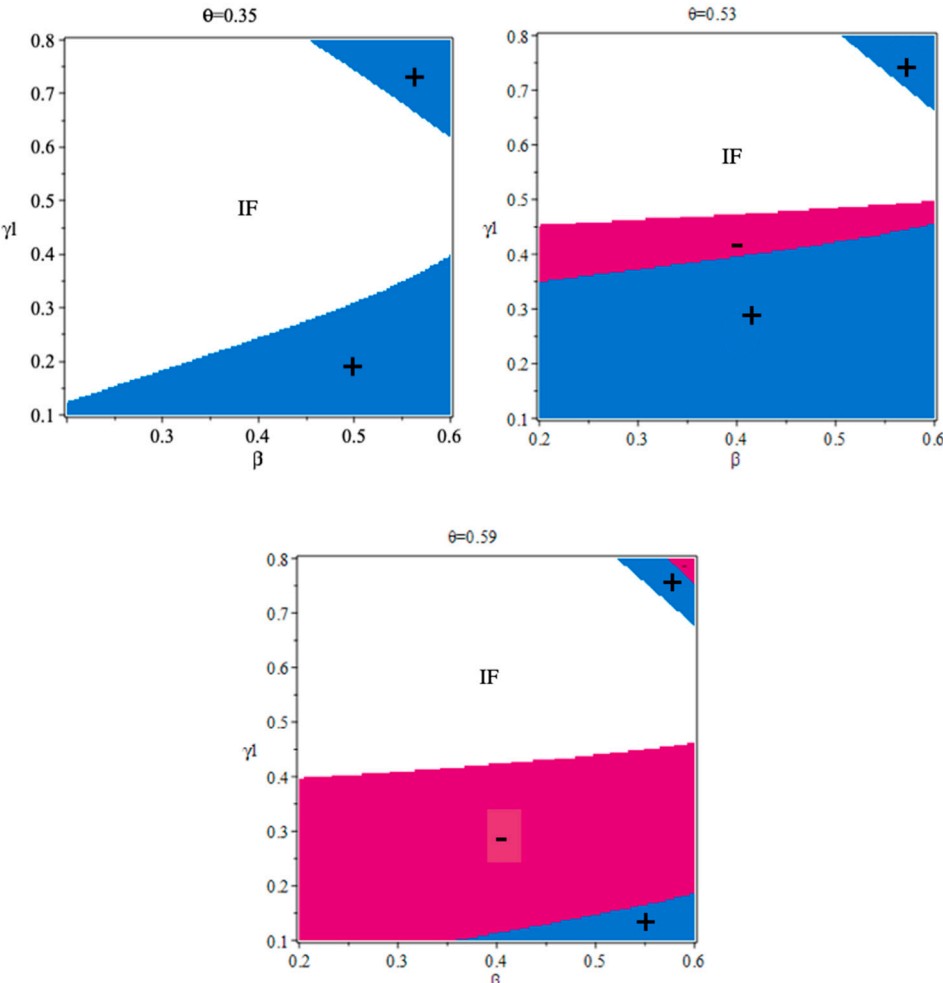

**Figure 8.** Comparison of the online price and offline prices of product 1 in scenario E. (Offline price–Online price).

In Scenario E, the online retail price for product 1 can be higher or lower than the offline price, depending on the variance of online market base ($\theta$). The online retail price for product 1 is lower than its counterpart offline price when the online market size is less than half. Generally speaking, the online retail price of product 1 is more likely to be higher than its offline retail price in the condition of a relatively large online market base.

The retailer aims at maximizing their total profits regarding the sale of the two complementary products. A high retail price of product 1 in Scenario E could lead to less demand for product 2 when the level of complementarity between two products is high. Consequently, the retailer tends to reduce the retail price for product 2 to prevent the loss of the demand. As the online market base increases, the retailer is forced to charge a lower offline price of product 1. To make up the profit loss in selling product 1, the retailer tends to raise the margin profit of product 2. Product 1's wholesale price is a strategic response to the pricing strategy of the retailer. The wholesale price of product 2 going down could be attributable to the entry of online channel for product 1. From corollary 3, based on the above assumption that $\gamma_1 = \gamma_0$, we found the demand for product 2 is not affected by the online market base. The added online channel has negative impacts on the demand for product 2 when the complementarity degree between the offline and online channels is the same as that between the two offline channels. Hence, the traditional manufacturer has the incentive to lower product 2's wholesale in order to increase their sales. Surprisingly, unlike the result in Zhao et al. (2017), product 1's online price was not always lower than its offline price in our study. This is due to the competition

pressure from the online sales of product 1. To the dominant retailer, the profit lost in the sale of product 1 can be compensated from increased margin profit and sales of product 2.

### 4.2. On Channel Demands

Now, before comparing the sales obtained from Scenarios N and E, we analyze the sensitivity of the demand for product 2 with reference to the parameter $\theta$, then we summarize the findings in the following corollary and claims.

**Corollary 3.** $\frac{\partial D_2^{E*}}{\partial \theta} = \frac{(a_1\beta - a_1)(\gamma_1 - \gamma_0)}{4\beta\gamma_0\gamma_1 + 8\beta^2 + 2\gamma_0^2 + 2\gamma_1^2 - 8}$ *(a) if* $\gamma_1 = \gamma_0$, *then* $\frac{\partial D_2^{E*}}{\partial \theta} = 0$; *(b) if* $\gamma_1 > \gamma_0$, *then* $\frac{\partial D_2^{E*}}{\partial \theta} > 0$; *(c) if* $\gamma_1 < \gamma_0$, *then* $\frac{\partial D_2^{E*}}{\partial \theta} < 0$.

**Claim 6.** *The decision to add the online channel results in reduced sales in both the offline channels of two manufacturers when the degree of complementarity between the traditional offline and online channels is the same as that between the two offline channels.*

**Claim 7.** *The integrated manufacturer's total sales (offline and online) can either be lower or higher in Scenario E than that in Scenario N for* $\theta \in (0.41, 0.6)$. *Otherwise, the integrated manufacturer's total sales (offline and online) will be higher after adding the online channel (Figure 9).*

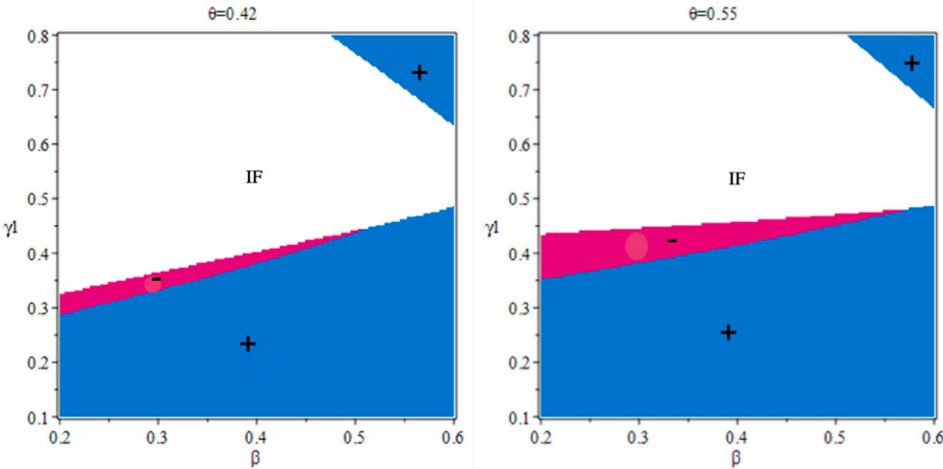

**Figure 9.** Compare the integrated manufacturer's total sales (sales in scenario E–sales in scenario N).

It is intuitive that adding the new online channel will reduce the sales of the integrated manufacturer's offline channel, because the online channel would take away some market share of product 1 from their offline channel. From corollary 3, we found the demand of product 2 is not affected by the online market base when the level of complementarity between the online and the retail channels is the same as that between the two retailer channels. The demand for product 2 also decreases as the online channel opens. The reason is that the demand for product 2 is inversely related to $\gamma_0$ and $\gamma_1$, which causes a decrease in the wholesale price of product 2 directly. In such a case, the ability to sell directly to consumers seems to be a hurt for the traditional manufacturer. Compared to Scenario N, now the consumers are offered the choice to either buy product 1 from the retailer or the integrated manufacturer. Consequently, the integrated manufacturer's total sales increases when their online market base is either relatively small or large. In contrast, when their online market base is comparable to their offline market base, the integrated manufacturer's total sales actually decrease in the case of low level of competition intensity between channels and relatively high level of complementarity between products. However, in general, adding the online channel enables the integrated manufacturer to increase sales volume.

### 4.3. On Channel Profits

In this section, channel members' profits are compared in Scenarios N and E, the findings are summed up as follows.

**Claim 8.** *The integrated manufacturer earns more when the online market base θ is relatively large, i.e., θ ∈ (0.63, 0.93). Otherwise, when θ is relatively small, the integrated manufacturer may earn either more or less depending on the level of complementarity between the two products, competition intensity, and the online market base (Figure 10).*

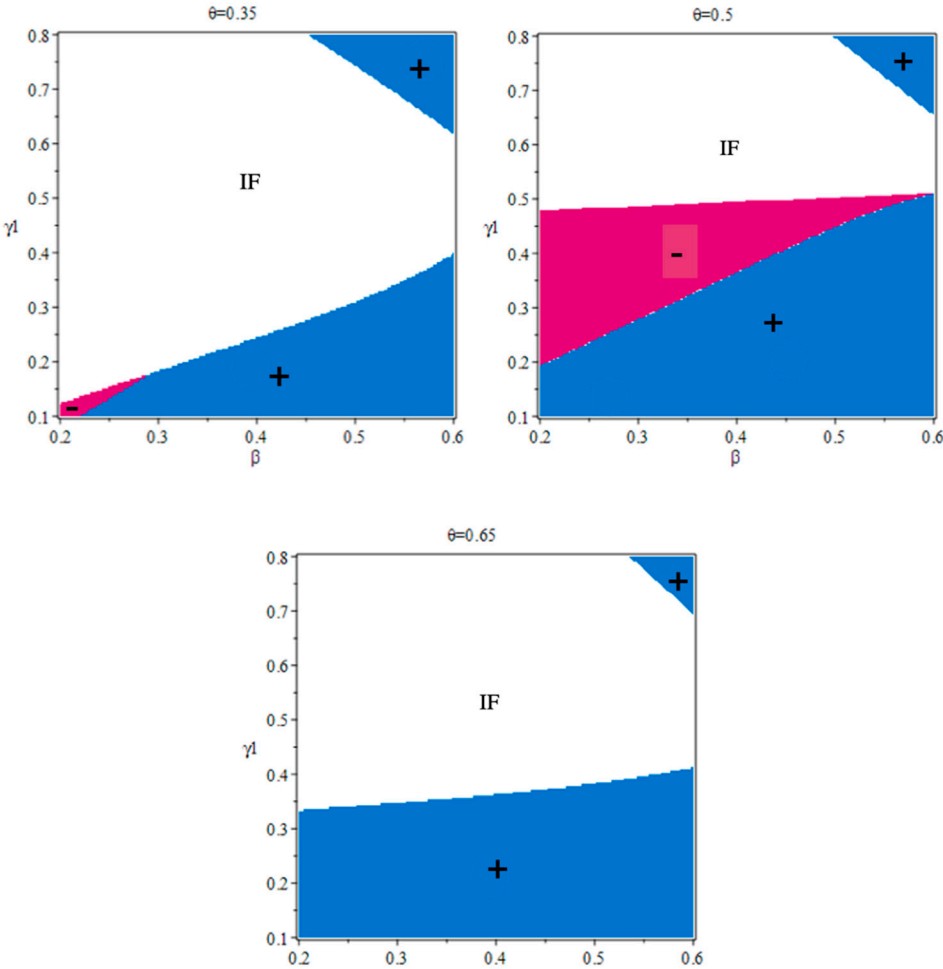

**Figure 10.** Compare the integrated manufacturer's profits (profit in scenario E–profit in scenario N).

Claim 8 suggests that the integrated manufacturer is willing to add an online channel when the online base $\theta$ is sufficiently large. For large values of $\theta$, i.e., $\theta \in (0.63, 0.93)$, the addition of a new online channel expands the total demand of the integrated manufacturer from the two channels. However, as the online market base $\theta$ is sufficiently small, the profit of the integrated manufacturer could be lower in Scenario E than that in Scenario N unless the complementarity level $\gamma_1$ or the competition intensity $\beta$ are relatively high. With increasing of the online market base $\theta$, the integrated manufacturer's profit could actually be lower in Scenario E than that in Scenario N when the complementarity level $\gamma_1$ is not too small. However, when the value of online market base $\theta$ continues to increase and exceeds an established threshold, the integrated manufacturer again earns more in the dual-channel.

**Claim 9.** *With the addition of a new online channel, the profit of the retailer generated by the sale of product 1 decreases, while their profit from selling product 2 may either increase or decrease, based on the values of parameters $\theta$, $\gamma_1$, and $\beta$. As a result, the addition of online channel reduces retailers' total revenue for both products. Furthermore, the traditional manufacturer, in general, earn less as the integrated manufacturer adds the online channel in the situation: the degree of complementarity between the traditional offline channel and online channel is the same as that between the two offline channels (Figure 11).*

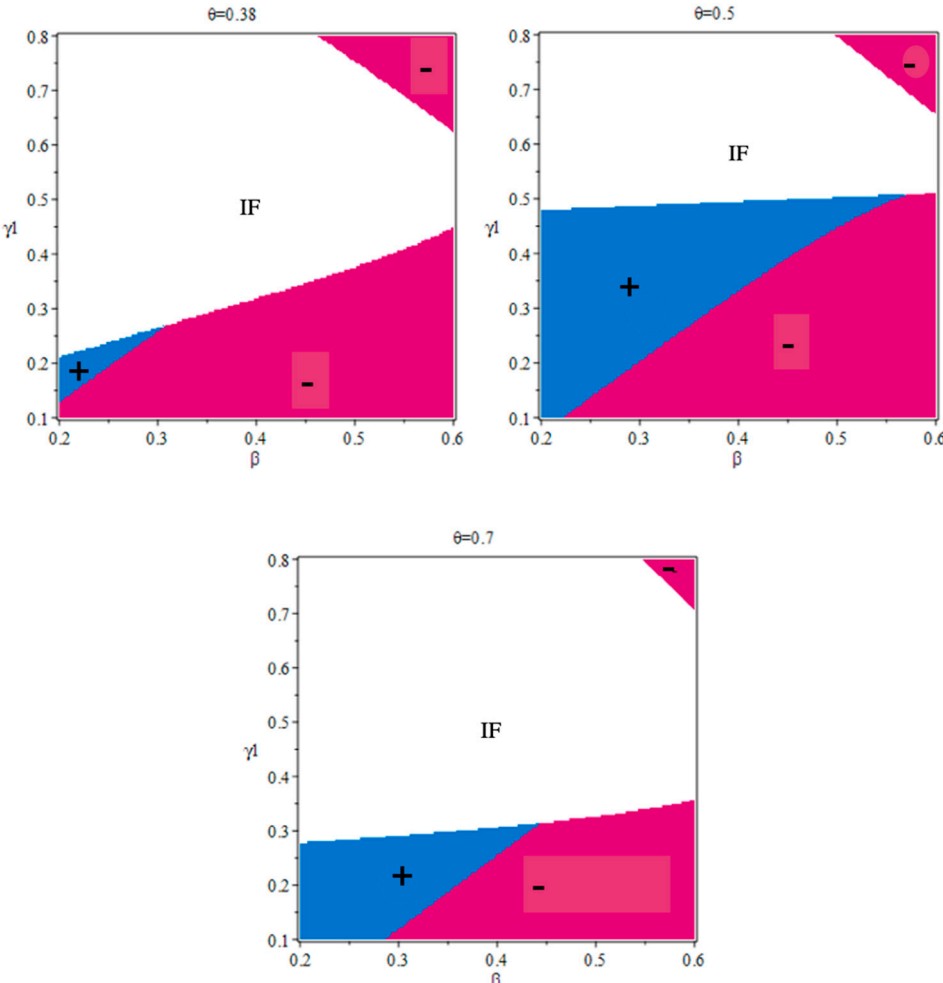

**Figure 11.** Compare the retailer's profits of product 2 (profit in scenario E–profit in scenario N).

As previously described in Claim 6, the offline demands of the two products decrease with adding the online channel. Consequently, the profit of the retailer from selling product 1 is actually lower after the addition of the new online channel. In view of this, the retailer, as the Stackelberg leader in pricing decisions, chooses to raise their profit margin of product 2 to compensate the profit loss in selling product 1. As a result, the retailer's profit generated from the sale of product 2 may increase. The conventional wisdom suggests that the increased competition with adding a new online channel may be hurt the retailer. It seems to be particularly pronounced in this setting, where the encroachment dissipates the power of the monopoly retailer. Before the addition of the online channel, the duopoly manufacturers have the same power in the market. From corollary 3, we found one might expect that an encroaching channel damages the market power of the traditional manufacturer when the degree of complementarity between the two retailer channels is not larger than that between the online and retail channels. Given that the sales of product 2 decrease in scenario E, the traditional manufacturer tends to charge a lower wholesale price in order to expand their demand. But, overall, the traditional manufacturer earns less in scenario E than in scenario N.

## 5. Extension

Until now, we have assumed that $\gamma_0$ is equal to $\gamma_1$. The main reason for this assumption was to make the analysis easier. Based on corollary 3, we found that the relationship between $\gamma_0$ and $\gamma_1$ affects the change of the demand for product 2. In this section, we extend our analysis to the circumstance that the degree of complementarity between product 1 in the online channel and product 2 in the offline channel is smaller than the level of complementarity between two products in the offline channels.

For ease of exposition, we set $\gamma_0 = \frac{1}{3}\gamma_1$ in the extended numercal study. Our analysis showed that when the online market base $\theta$ is small enough, the integrated manufacturer tends not to add the online channel unless the level of complementarity ($\gamma_1$) between the two offline channels is relatively low. Under such a situation, as shown in Figure 12, the feasible domain expands significantly and covers a wider range of $\gamma_1$. Otherwise, the feasible domain remains comparable to that in the previous study when $\gamma_0 = \gamma_1$.

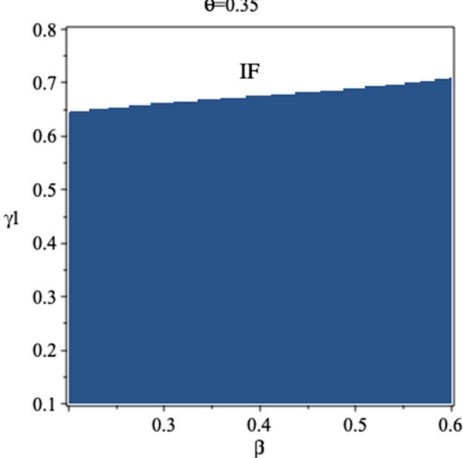

**Figure 12.** Feasible domain for the scenario E ($\gamma_0 = \frac{1}{3}\gamma_1$).

There is more room for the integrated manufacturer to make more profits when adding the online channel as the feasible domain expands. Meanwhile, the monopolistic retailer is still worse off when the online channel is added. However, we found that the earning of the traditional manufacturer is different from the previous result. After the addition of online channel, the traditional manufacturer could earn more when the competition intensity between offline and online channels is relatively small. From corollary 3, we found the demand of product 2 and the online base are positively correlated when the degree of complementarity ($\gamma_1$) between two offline channels is larger than the degree of complementarity ($\gamma_0$) between the offline and the online channels; this is due to the rising competition intensity between product 1 from two different channels. As shown in Figure 8, the online price of product 1 is more likely to be higher than its offline retail price for relatively large online market base, so the low degree of complementarity between the online channel and offline channel drives the demand of product 2 to increase when the competition intensity is fairly small (see Figure 13), and consequently, the profit of traditional manufacturer also increases. One might expect that an encroaching channel makes for the market power of the traditional manufacturer under certain conditions.

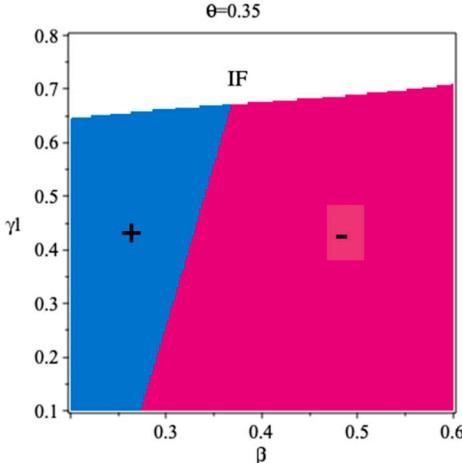

**Figure 13.** Compare the traditional manufacturer's sale (sale in scenario E–sale in scenario N) ($\gamma_0 = \frac{1}{3}\gamma_1$).

## 6. Conclusions

This paper considered a supply chain that the two manufacturers sell green complementary products to a dominant offline retailer, who leads the green supply chain as a core member. We investigated whether a manufacturer (the integrated manufacturer) should add an online channel by analyzing and comparing the following two models: (1) Scenario N: the benchmark model where the integrated manufacturer does not add an online channel; (2) Scenario E: the dual-channel model where the integrated manufacturer adds an online channel. We formulated power structure as a retailer-Stackelberg model and analyzed channel members' pricing decisions. The main contributions are summarized as follows. First, supply chain members should strive to make the degree of complementarity between online and offline for the two green complementary products lower than that between the offline channels, which will benefit the supply chain members. Second, when the online and offline market bases of the green complementary products are almost same and the level of complementarity is moderate, the integrated manufacturer does not add the online channel, and the traditional supply chain may bring more revenue.

We determined under what conditions it is a good choice for the integrated manufacturer to add the online channel. Due to mathematical tractability, we referred to numerical studies to characterize the optimal solutions and derive meaningful insights. For ease of exposition, we set the degree of complementarity between product 1 in the online channel and product 2 in the offline channel equal to that between two green complementary products in the retail channels. Not all values of parameters satisfied the necessary conditions for ensuring channel members' participation. Our analysis showed that the integrated manufacturer is willing to add the online channel as the online market base is relatively large, however, the integrated manufacturer's profit could be lower in scenario E than that in scenario N, as the online market base is fairly small. What is more, the integrated manufacturer would not add the online channel when online and offline market bases are comparable and the level of complementarity is moderate. The retailer's profit generated by the sale of product 1 decreases with adding an online channel, while its profit from selling product 2 may either increase or decrease. This result is congruent with the view that adding online channel by a manufacturer undermines the offline retailer's channel power [33]. The profits of the retailer come from both green complementary products. The integrated manufacturer competes directly with the retailer by the online channel. Consequently, the profit of the dominant retailer from selling product 1 is actually lower after the addition of a new online channel. The retailer, as the Stackelberg leader, can exercise their power to raise her profit margin of product 2 to compensate the profit loss in selling product 1. The added online channel damages the market power of the traditional manufacturer when the degree of complementarity between the two offline channels is not larger than the degree of complementarity between the online

channel and offline channel, and as a result, the traditional manufacturer tends to charge a lower wholesale price in order to expand their demand.

In the extended study, we found that the smaller degree of complementarity between the online channel and offline channel benefits the duopolistic manufacturers. In particular, the traditional manufacturer could earn more, as the competition intensity between the offline and online channels is relatively small. What is more, with the demand of product 2 increasing, the retailer will have more total sales of two products. Our results provide several managerial implications. First, it is beneficial for the retailer to balance the online and offline market bases of product 1 by improving the sales environment of the physical store, in such a situation, the integrated manufacturer would have less incentive to add the new online channel, and hence, the retailer's profit will not be damaged. Second, the integrated manufacturer can vary their marketing actions for decreasing complementarity level between the online and retail channels to get support from the traditional manufacturer. Lastly, the traditional manufacturer can take steps to increase the degree of complementarity between the two offline channels, aiming at reducing the complementarity of the two green products between the online and offline channels.

We can extend this paper in several aspects in the future. First, this study assumed that the demand is a linear function of certain parameters and deterministic. However, in reality, random variations often occur. Incorporation of randomness into the model will make the analysis very complicated but merits further exploration. Second, in this paper the two manufacturers had symmetrical power, but in the real market, there is power asymmetry between manufacturers. Therefore, it is interesting to examine how power asymmetry would impact channel members' decisions. Third, we supposed that the monopoly retailer and the two manufacturers are risk neutral. In future studies, we can analyze the effect of risk aversion on optimal decisions and pricing strategies of channel members. Lastly, a bundle sale of complementary products by the retailer can be another possible direction for future studies.

**Author Contributions:** Q.G. wrote the manuscript, X.Y. helped perform the analysis with constructive discussions, and B.L. masterminded this research and contributed significantly to analysis. All authors have read and agreed to the published version of the manuscript.

**Funding:** This research was funded by National Natural and Science Foundation of China with Grant Number 71971134, Science and Technology Ministry of China for Cruise Program with Grant Number 2018-473 and Ministry of Education of China for Humanities and Social Sciences Foundation with Grant Number 18YJA630143.

**Acknowledgments:** The authors gratefully acknowledge the support from National Natural and Science Foundation of China, Science and Technology Ministry of China for Cruise Program with and Ministry of Education of China for Humanities and Social Sciences Foundation.

**Conflicts of Interest:** The authors declare no conflict of interest.

## Appendix A

$$A = \begin{vmatrix} -4 - 2\gamma_1^2 & -6\gamma_1 \\ -6\gamma_1 & -4 - 2\gamma_1^2 \end{vmatrix} \tag{A1}$$

$$A_1 = \begin{vmatrix} -c_1 - \gamma_1 c_2 - 3a_1 - 2\gamma_1 a_2 & -6\gamma_1 \\ -c_2 - \gamma_1 c_1 - 3a_2 - 2\gamma_1 a_1 & -4 - 2\gamma_1^2 \end{vmatrix} \tag{A2}$$

$$A_2 = \begin{vmatrix} -4 - 2\gamma_1^2 & -c_1 - \gamma_1 c_2 - 3a_1 - 2\gamma_1 a_2 \\ -6\gamma_1 & -c_2 - \gamma_1 c_1 - 3a_2 - 2\gamma_1 a_1 \end{vmatrix} \tag{A3}$$

$$\begin{pmatrix} B_{11} & B_{12} & B_{13} \\ B_{21} & B_{22} & B_{23} \\ B_{31} & B_{32} & B_{33} \end{pmatrix} = \begin{pmatrix} \frac{1}{\beta^2-1} & \frac{\beta}{\beta^2-1} & 0 \\ \frac{\beta}{2\beta^2-2} & \frac{1}{2\beta^2-2} & 0 \\ \frac{-\beta\gamma_0}{2\beta^2-2} & \frac{-\gamma_1}{2\beta^2-2} & -1 \end{pmatrix} \begin{pmatrix} 1 & \gamma_1 & -c_1 + (c_1+\delta)\beta - (1-\theta)a_1 \\ -\beta & \gamma_0 & -c_1 - \delta + c_1\beta - \theta a_1 \\ \gamma_1 & 1 & -c_2 - a_2 \end{pmatrix}$$

$$c_{11} = 2(-B_{11}+1)(\beta B_{21}-1) - 2B_{31}(-\gamma_0 B_{21} - \gamma_1)$$

$$c_{12} = (-B_{11}+1)(\beta B_{22}-\gamma_1) - B_{12}(\beta B_{21}-1) - B_{31}(-\gamma_0 B_{22}-1) + (-B_{32}+1)(-B_{21}\gamma_0 - \gamma_1)$$

$$c_{13} = (B_{11}-1)((1-\theta)a_1 + B_{23}\beta) + B_{13}(\beta B_{21}-1) + B_{31}(-B_{23}\gamma_0 + a_2) + B_{33}(-B_{21}\gamma_0 - \gamma_1)$$

(A4)

$$c_{21} = -B_{12}(\beta B_{21}-1) + (-B_{11}+1)(\beta B_{22}-\gamma_1) + (-B_{32}+1)(-B_{21}\gamma_0 - \gamma_1) - B_{31}(-\gamma_0 B_{22}-1)$$

$$c_{22} = -2B_{12}(\beta B_{22}-\gamma_1) + 2(-B_{32}+1)(-B_{22}\gamma_0-1)$$

$$c_{23} = B_{12}((1-\theta)a_1 + B_{23}\beta) + B_{13}(\beta B_{22}-\gamma_1) - (-B_{32}+1)(-B_{23}\gamma_0 + a_2) + B_{33}(-B_{22}\gamma_0 - 1)$$

**Appendix B**

**Proof of Proposition 1.** From Equations (1)–(4), we write the first-order partial derivatives of $\pi_{m1}^N(w_1)$ with reference to $w_1$, and $\pi_{m2}^N(w_2)$ to $w_2$ by

$$\frac{\delta\pi_{m1}^N(w_1)}{\delta w_1} = -p_1 - p_2\gamma_1 + a_1 - (w_1 - c_1),$$ (A5)

$$\frac{\delta\pi_{m2}^N(w_2)}{\delta w_2} = -p_2 - p_1\gamma_1 + a_2 - (w_2 - c_2).$$ (A6)

We then calculate the second-order derivatives of $\pi_{m1}^N(w_1)$ and $\pi_{m2}^N(w_2)$ with respect to $w_1$ and $w_2$ as

$$\frac{\partial^2\pi_{m1}^N(w_1)}{\partial w_1^2} = -1 \text{ and } \frac{\partial^2\pi_{m2}^N(w_2)}{\partial w_2^2} = -1.$$ (A7)

From Equation (A7), we find that $\pi_{mi}^N(w_i)$ is concave with $w_i$, I = 1,2. As a result, setting Equations (A5) and (A6) to zero, at the same time solving them, we obtain Equations (14) and (15). From Equations (1), (2), (14), (15) and (5), the first–order partial derivatives of $\pi_r^N(p_1, p_2)$ with reference to $p_1$ and $p_2$ can be shown as

$$\frac{\delta\pi_r^N(p_1, p_2)}{\delta p_1} = (-4 - 2\gamma_1^2)p_1 - 6\gamma_1 p_2 + c_1 + \gamma_1 c_2 + 3a_1 + 2\gamma_1 a_2,$$ (A8)

$$\frac{\delta\pi_r^N(p_1, p_2)}{\delta p_2} = -6\gamma_1 p_1 + (-4 - 2\gamma_1^2)p_2 + c_2 + \gamma_1 c_1 + 3a_2 + 2\gamma_1 a_1,$$ (A9)

and the Hessian matrix

$$H_1 = \begin{vmatrix} -4 - 2\gamma_1^2 & -6\gamma_1 \\ -6\gamma_1 & -4 - 2\gamma_1^2 \end{vmatrix}$$ (A10)

is negative definite when $-4-2\gamma_1^2 < 0$ and $4\gamma_1^4 - 20\gamma_1^2 + 16 > 0$. It is obvious that the two conditions are satisfied for $0 < \gamma_1 < 1$. Therefore, $\pi_r^N(p_1, p_2)$ is jointly concave with $p_1$ and $p_2$. Setting Equations (A8) and (A9) to zero, at the same time solving them, Equation (16) is obtained. Some conditions need to be satisfied to obtain nonnegative profits and positive pricing decisions, namely, $w_i^{N*} > c_i$ and $p_i^{N*} > w_i^{N*}, i = 1, 2$. All conditions are verified when parameter values satisfy the condition

$-c_1\gamma_1^2 + \gamma_1(a_2 + c_2) - 2a_1 + 2c_1 < 0, -c_2\gamma_1^2 + \gamma_1(a_1 + c_1) - 2a_2 + 2c_2 < 0, -c_1\gamma_1^2 + \gamma_1 a_2 - a_1 + c_1 < 0$, and $-c_2\gamma_1^2 + \gamma_1 a_1 - a_2 + c_2 < 0$. Thus, we prove Proposition 1. □

**Proof of Proposition 2.** From Equations (6)–(10), we derive the first–order partial derivatives of $\pi_{m1}^E(w_1, p_0)$ with reference to $w_1$ and $p_0$, and $\pi_{m2}^E(w_2)$ to $w_2$ as follows

$$\frac{\delta\pi_{m1}^E(w_1, p_0)}{\delta w_1} = -w_1 + 2\beta p_0 - p_1 - \gamma_1 p_2 + (1 - \theta)a_1 + c_1 - (c_1 + \delta)\beta,) \tag{A11}$$

$$\frac{\delta\pi_{m1}^E(w_1, p_0)}{\delta p_0} = \beta w_1 - 2p_0 + \beta p_1 - \gamma_0 p_2 - \beta c_1 + \theta a_1 + c_1 + \delta, \tag{A12}$$

$$\frac{\delta\pi_{m2}^E}{\delta w_2} = -\gamma_0 p_0 - w_2 - \gamma_1 p_1 - p_2 + a_2 + c_2, \tag{A13}$$

and the Hessian matrix:

$$H_2 = \begin{vmatrix} -1 & 2\beta \\ \beta & -2 \end{vmatrix}, \tag{A14}$$

is negative definite when the condition $2 - 2\beta^2 > 0$ holds. Using the assumption that the cross-price sensitivity is smaller than own price sensitivity ($0 < \beta < 1$), we know that $\pi_{m1}^E$ is jointly concave with $w_1$ and $p_0$, and $\pi_{m2}^E$ is concave with $w_2$. Then, Setting Equations (A11)–(A13) to zero, at the same time solving them, Equation (21) are obtained. As a result, Proposition 2 is proved. □

**Proof of Proposition 3.** We can write the first–order partial derivatives of $\pi_r^E(p_1, p_2)$ with reference to $p_1$ and $p_2$ from Equations (7), (8), (11), and (21) as follows

$$\frac{\delta\pi_r^E(p_1, p_2)}{\delta p_1} = c_{11}p_1 + c_{12}p_2 - c_{13} \tag{A15}$$

$$\frac{\delta\pi_r^E(p_1, p_2)}{\delta p_1} = c_{21}p_1 + c_{22}p_2 - c_{23}, \tag{A16}$$

and the Hessian matrix:

$$H_4 = \begin{vmatrix} c_{11} & c_{12} \\ c_{21} & c_{22} \end{vmatrix}, \tag{A17}$$

is negative definite when parameter values satisfy the condition:

$$\left(\beta^2\gamma_1^2 - 2\beta\gamma_0\gamma_1 - 2\beta^2 - \gamma_0^2 - 2\gamma_1^2 + 2\right)\left(2\beta\gamma_0\gamma_1 + 4\beta^2 + \gamma_0^2 + 2\gamma_1^2 - 4\right) < 0. \tag{A18}$$

So, $\pi_r^E(p_1, p_2)$ is jointly concave in $p_1$ and $p_2$. Setting Equations (A15) and (A16) to zero, at the same time solving them yields Equation Equation (22). Thus, Proposition 5 is proved. □

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
