# Peer review of "Pricing Decisions on Online Channel Entry for Complementary Products in a Dominant Retailer Supply Chain"

_sustainability, doi:10.3390/su12125007_

Round 1
Reviewer 1 Report
It is my great honor to be appointed a reviewer of the paper submitted to Sustainability. I see the paper, finding that the paper is interesting.
Author Response
Thank you very much for your appreciation of our work, the paper is checked by English editor again.
Reviewer 2 Report
The paper is not relevant to the Journal Sustainability. This paper not involves sustainability and sustainable development. This paper in actual version, in reviewer opinion, has not satisfactory standard for publication.
Author Response
Point 1: The paper is not relevant to the Journal Sustainability. This paper not involves sustainability and sustainable development.
Response 1: Thanks for your reminding. We are very sorry for our negligence of research background. Through our subsequent investigations, we find that green complementary products are becoming more and more popular in the market, for example, a practical case of green products is about the house decoration business, when Noritz sells an energy-saving water heater, Chicago Furnace also provides environment friendly faucets with water-saving outlet options. At the same time, the manufacturers of green complementary products may confront the same channel selection and pricing problems in the supply chain. manufacturers are facing the same channel selection and pricing problems. What’s more, we find a paper “To Cooperate or Not? An Analysis of Complementary Product Pricing in Green Supply Chain” on the pricing problem of green complementary products, so we reselect the green complementary products as the background of the revised paper.
Green production has gradually become a popular and growing field due to its important role in economic and environmental sustainability. Some researchers focus on the issue of green product to reduce environment damage while maintaining profits (Tsai et al., 2016). Now, as the laws and regulations become more and more rigorous, the organizations can’t ignore product green level. For example, the company Apple, as core designer of the product, it sets the green product target, then for satisfy the target its suppliers will increase the recycling of manufacture material and reduce the use of toxic metal. This paper mainly considers the sales of green complementary products.
For example, Hewlett-Packard (HP) is in the leading position in laser printers; what’s more, its components are provided for the same target market by Canon (Lewis Jordan,1999). The green complementary products refer to environmentally friendly complementary products, for example, a practical case of green products is about the house decoration business, when Noritz sells an energy-saving water heater, Chicago Furnace also provides environment friendly faucets with water-saving outlet options (Contractor Magazine. Oct 2017, Vol. 64 Issue 10, p43.). Therefore, the manufacturers of green complementary products may confront the problem of channel selection and pricing in the supply chain. Recently, complementary products are sold by more and more manufacturers through dual channel in a supply chain.
Point 2: This paper in actual version, in reviewer opinion, has not satisfactory standard for publication.
Response 2: Thanks for your reminding. We have made correction according to the Reviewer’s comments. First, we improved the introduction part, some sentences added to show more on motivation, and more of background examples about green complementary products, what’s more, the relevant references are added in the introduction. Second, the authors re-summarize the findings,and two unique contributions added in the revised paper. Last, we add a hint after the Propositions to make the paper more clear. Besides, the English language is checked by English editor again.
In such instances, the direct channel of manufacturer competes with the traditional channel of the retailer, as a result, the traditional retailer ordinarily complains the demands that are satisfied through the manufacturer’s direct channel should belong to the retailer’s traditional channel (Chiang et al., 2003). This may result in the channel conflict, which weakens the attempts to build the cooperative relationship for channel members (Li et al., 2016). Sales through manufacturer’s direct channel cannibalize market share and profits of the traditional retailer dramatically, some operational strategies have been adopted (such as providing services and value-added products, etc) by a lot of traditional retailers to strengthen their core competitiveness. As reported by the New York Times, there are roughly 42% of the manufacturers reconstruct the traditional channel structures through adding the direct online sales to satisfy different customer segments that is not reachable by the traditional channel, which are generating the combination channel of the traditional channel and booming direct online channel (also called the dual-channel).
However, not all studies have found that the addition of direct channel creates conflicts, some researchers show that the addition new channel is beneficial to other channel members actually. For example, Arya et al. (2007) demonstrate the bright side of manufacturer encroachment. Therefore, on this basis, we study the interesting issues whether a manufacturer open a new direct channel in the supply chain, where green complementary products are sold, and how other channel members adopt strategies when adding an online channel into the traditional channel.
Green production has gradually become a popular and growing field due to its important role in economic and environmental sustainability. Some researchers focus on the issue of green product to reduce environment damage while maintaining profits (Tsai et al., 2016). Now, as the laws and regulations become more and more rigorous, the organizations can’t ignore product green level. For example, the company Apple, as core designer of the product, it sets the green product target, then for satisfy the target its suppliers will increase the recycling of manufacture material and reduce the use of toxic metal. This paper mainly considers the sales of green complementary products.
For example, Hewlett-Packard (HP) is in the leading position in laser printers; what’s more, its components are provided for the same target market by Canon (Lewis Jordan,1999). The green complementary products refer to environmentally friendly complementary products, for example, a practical case of green products is about the house decoration business, when Noritz sells an energy-saving water heater, Chicago Furnace also provides environment friendly faucets with water-saving outlet options (Contractor Magazine. Oct 2017, Vol. 64 Issue 10, p43.). Therefore, the manufacturers of green complementary products may confront the problem of channel selection and pricing in the supply chain. Recently, complementary products are sold by more and more manufacturers through dual channel in a supply chain.
the integrated manufacturer represents the manufacturer that can sell the product through two channels: the direct online channel and traditional offline channel, while the traditional manufacturer mean that the manufacturer only sell product through the traditional offline channel.
Third, Supply chain members should strive to make the degree of complementarity between online and offline for the two products lower than that between the offline channels, which will benefit the supply chain members. Fourth, when the online and offline market bases are almost same and the level of complementarity is moderate, the integrated manufacturer do not add the online channel, and the traditional supply chain may bring more revenue.
(Proofs are shown in Appendix B).
The proof of Proposition 1 and other remaining proofs appear in Appendix B. According to the above equations, we can obtain the results of Corollary 1.
Special thanks to you for your good comments. Those comments are all valuable and very helpful for revising and improving our paper, as well as the important guiding significance to our researches. We appreciate for your warm work earnestly, and hope that the correction will meet with approval.
Once again, thank you very much for your comments and suggestions.
Reviewer 3 Report
The paper handles an interesting topic. However, introduction part is somewhat weak. Authors should elaborate more on motivation, show more of background examples, and give clearer definitions of concepts. For example, how is an integrated manufacturer differentiated from a traditional manufacturer?
At the end of literature review, authors suggest two contributions of the paper. But both of them are not unique contribution of this paper. But it seems too general. There are existing studies that claimed the similar issues. Therefore, it would be great if authors can either how this paper is different from those previous works or bring up and add a unique contribution in which readers can't find anywhere else.
Propositions are not explained fully. I want authors to add more description of the math results (equations) that are shown up in propositions.
Author Response
Point 1: The paper handles an interesting topic. However, introduction part is somewhat weak. Authors should elaborate more on motivation, show more of background examples, and give clearer definitions of concepts. For example, how is an integrated manufacturer differentiated from a traditional manufacturer?
Response 1: Thanks for your reminding. We have improved this part according to your suggestions. For introduction part new sentences added to show more on motivation, and more of background examples about green complementary products, what’s more, the specific definitions of the integrated manufacturer and the traditional manufacturer are added in the introduction.
In such instances, the direct channel of manufacturer competes with the traditional channel of the retailer, as a result, the traditional retailer ordinarily complains the demands that are satisfied through the manufacturer’s direct channel should belong to the retailer’s traditional channel (Chiang et al., 2003). This may result in the channel conflict, which weakens the attempts to build the cooperative relationship for channel members (Li et al., 2016). Sales through manufacturer’s direct channel cannibalize market share and profits of the traditional retailer dramatically, some operational strategies have been adopted (such as providing services and value-added products, etc) by a lot of traditional retailers to strengthen their core competitiveness. As reported by the New York Times, there are roughly 42% of the manufacturers reconstruct the traditional channel structures through adding the direct online sales to satisfy different customer segments that is not reachable by the traditional channel, which are generating the combination channel of the traditional channel and booming direct online channel (also called the dual-channel).
However, not all studies have found that the addition of direct channel creates conflicts, some researchers show that the addition new channel is beneficial to other channel members actually. For example, Arya et al. (2007) demonstrate the bright side of manufacturer encroachment. Therefore, on this basis, we study the interesting issues whether a manufacturer open a new direct channel in the supply chain, where green complementary products are sold, and how other channel members adopt strategies when adding an online channel into the traditional channel.
Green production has gradually become a popular and growing field due to its important role in economic and environmental sustainability. Some researchers focus on the issue of green product to reduce environment damage while maintaining profits (Tsai et al., 2016). Now, as the laws and regulations become more and more rigorous, the organizations can’t ignore product green level. For example, the company Apple, as core designer of the product, it sets the green product target, then for satisfy the target its suppliers will increase the recycling of manufacture material and reduce the use of toxic metal. This paper mainly considers the sales of green complementary products.
For example, Hewlett-Packard (HP) is in the leading position in laser printers; what’s more, its components are provided for the same target market by Canon (Lewis Jordan,1999). The green complementary products refer to environmentally friendly complementary products, for example, a practical case of green products is about the house decoration business, when Noritz sells an energy-saving water heater, Chicago Furnace also provides environment friendly faucets with water-saving outlet options (Contractor Magazine. Oct 2017, Vol. 64 Issue 10, p43.). Therefore, the manufacturers of green complementary products may confront the problem of channel selection and pricing in the supply chain. Recently, complementary products are sold by more and more manufacturers through dual channel in a supply chain.
the integrated manufacturer represents the manufacturer that can sell the product through two channels: the direct online channel and traditional offline channel, while the traditional manufacturer mean that the manufacturer only sell product through the traditional offline channel.
Point 2: At the end of literature review, authors suggest two contributions of the paper. But both of them are not unique contribution of this paper. But it seems too general. There are existing studies that claimed the similar issues. Therefore, it would be great if authors can either how this paper is different from those previous works or bring up and add a unique contribution in which readers can't find anywhere else.
Response 2: Thanks for your reminding. We have added two unique contributions at the end of literature review in the revised paper.
Third, Supply chain members should strive to make the degree of complementarity between online and offline for the two products lower than that between the offline channels, which will benefit the supply chain members. Fourth, when the online and offline market bases are almost same and the level of complementarity is moderate, the integrated manufacturer do not add the online channel, and the traditional supply chain may bring more revenue.
Point 3: Propositions are not explained fully. I want authors to add more description of the math results (equations) that are shown up in propositions.
Response 3: Thanks for your reminding. We are very sorry for our negligence of detailed explanation of Propositions, we place the specific explanation in the Appendix B. In order to make more clear, a hint added after the Proposition 1.
(Proofs are shown in Appendix B).
The proof of Proposition 1 and other remaining proofs appear in Appendix B. According to the above equations, we can obtain the results of Corollary 1.
Special thanks to you for your good comments. We tried our best to improve the manuscript and made some changes in the manuscript. These changes will not influence the content and framework of the paper. And the changes marked in red in revised paper.We appreciate for your warm work earnestly, and hope that the correction will meet with approval.
Once again, thank you very much for your comments and suggestions.
Round 2
Reviewer 2 Report
Please complete the conclusion for green complementary products.
Author Response
Point 1: Please complete the conclusion for green complementary products.
Response 1: Thanks for your reminding. We have made correction according to the Reviewer’s comments. We add the conclusion about the green complementary products.
This paper considers a supply chain that the two manufacturers sell green complementary products to a dominant offline retailer, who leads the green supply chain as a core member. We investigate whether a manufacturer (the integrated manufacturer) should add an online channel by analyzing and comparing the following two models: (1) Scenario N: the benchmark model where the integrated manufacturer does not add an online channel; (2) Scenario E: the dual-channel model where the integrated manufacturer adds an online channel. We formulate power structure as a retailer-Stackelberg model and analyze channel members’ pricing decisions. The main contributions are summarized as follows. First, Supply chain members should strive to make the degree of complementarity between online and offline for the two green complementary products lower than that between the offline channels, which will benefit the supply chain members. Second, when the online and offline market bases of the green complementary products are almost same and the level of complementarity is moderate, the integrated manufacturer do not add the online channel, and the traditional supply chain may bring more revenue.
Special thanks to you for your good comments. Those comments are all valuable and very helpful for revising and improving our paper, as well as the important guiding significance to our researches. We appreciate for your warm work earnestly, and hope that the correction will meet with approval.
Once again, thank you very much for your comments and suggestions.